# Closing the Knowledge Gap in the Long-Term Health Effects of Natural Disasters: A Research Agenda for Improving Environmental Justice in the Age of Climate Change

**DOI:** 10.3390/ijerph192215365

**Published:** 2022-11-21

**Authors:** Arnab K. Ghosh, Martin F. Shapiro, David Abramson

**Affiliations:** 1Department of Medicine, Weill Cornell Medical College, Cornell University, Ithaca, NY 10065, USA; 2School of Global Public Health, New York University, 715/719 Broadway 12th Floor Room 1214, New York, NY 10003, USA

**Keywords:** climate change, natural disasters, structural disadvantage, environmental justice, transdisciplinary

## Abstract

Natural disasters continue to worsen in both number and intensity globally, but our understanding of their long-term consequences on individual and community health remains limited. As climate-focused researchers, we argue that a publicly funded research agenda that supports the comprehensive exploration of these risks, particularly among vulnerable groups, is urgently needed. This exploration must focus on the following three critical components of the research agenda to promote environmental justice in the age of climate change: (1) a commitment to long term surveillance and care to examine the health impacts of climate change over their life course; (2) an emphasis on interventions using implementation science frameworks; (3) the employment of a transdisciplinary approach to study, address, and intervene on structural disadvantage among vulnerable populations. Without doing so, we risk addressing these consequences in a reactive way at greater expense, limiting the opportunity to safeguard communities and vulnerable populations in the era of climate change.

## 1. Introduction

The last two years have revealed the destructive power of extreme weather events. Driven by the effects of a changing climate, devastating heat waves across the Indian subcontinent and Europe, the hurricane season in the southern United States, and dramatic flooding of over one-third of Pakistan are only some examples of the havoc caused [1]. This is not an anomaly. Natural disasters continue to worsen in both number and intensity in the US and also globally [2]. However, our understanding of their long-term consequences on individual and community health remains limited. The objective of this commentary is to articulate the urgent need for a publicly funded research agenda that examines these risks, and how to address the disproportionate risk of climate change-related morbidity and mortality among vulnerable patient populations. Such evidence can be used to improve existing national disaster planning frameworks that currently emphasize short-term post-recovery efforts, bringing a clearer emphasis on vulnerable populations. Furthermore, comprehensive evidence about the long-term trajectory of adverse health outcomes for victims of climate change-amplified events will enable healthcare professionals and policymakers globally to better identify at-risk individuals and develop interventions to prevent associated morbidity and mortality, particularly among those groups most vulnerable to climate change-related impacts, including children, older adults, socioeconomically disadvantaged groups, and minorities—a key tenet of environmental justice [3]. 

## 2. Limited Understanding Despite Increasing Events

Natural disasters, such as hurricanes, cyclones, flooding, and wildfires, have the potential to cause long-term disruption to their victims’ health, particularly in the era of climate change [4,5]. Property damage, limited access to healthcare [6], and economic disruption in affected areas [7] have been shown to increase the risk of displacement and place victims at risk of adverse health outcomes. Post-disaster studies have described increased rates of respiratory disease, heart disease, and stroke [8], as well as anxiety/depression [9], and demonstrate changes to healthcare costs, utilization, and rates of certain diseases upwards of 12 months after some extreme weather events [10]. Specifically, some studies suggest that risk from a broad range of cardiovascular diseases, such as myocardial infarction, stroke, and heart failure, remains higher than normal beyond 1 year after hurricane landfall for the general population but, particularly, for more vulnerable subgroups, such as the elderly, low-income individuals, and those who are unemployed [11,12,13,14,15]. In the United States, this focus on short-term recovery is reflected in policy frameworks, such as the Federal Emergency Management Agency’s National Disaster Recovery Framework, disaster-related Medicaid insurance waivers and expansions, Stafford Act funding guidance, and disaster-specific Community Development Block Grant funding mechanisms, most of which focus on the first 36 months after the event [16]. These frameworks emphasize the restoration of short-term emergency healthcare access in the first months after the event to address the immediate health needs resulting from natural disasters (e.g., injuries directly from the disaster, infections from disruption to water sources and exposure, medication needs) [16,17]. Informed by these short-term perspectives, authorities have historically focused health-related resources on addressing the physical environment (e.g., reopening hospitals and clinics), thus, prioritizing emergency care over management of chronic non-communicable diseases including cardiac disease, diabetes, and mental health [18]. 

As a result, the longer-term health consequences resulting from natural disasters remain understudied and underemphasized in disaster recovery planning despite the likelihood of their increased frequency driven by climate change. Few studies have followed patient cohorts over longer timeframes, likely influenced by methodological challenges in attributing downstream health impacts to extreme weather events [19]. This, in turn, limits the ability of recovery experts and governmental and public health officials to appropriately identify populations at risk of the enduring impacts of climate change-amplified extreme weather events [20], to consider the mechanisms by which disaster disruptions and stressors affect population health, and to allocate resources to mitigate these downstream health impacts. Even estimating disaster-related deaths remains a contentious and disputed arena, a point underscored in the US National Academies of Science, Engineering, and Medicine’s report *A Framework for Assessing Mortality and Significant Morbidity After Large-Scale Disasters* [17]. Published in 2020, the report calls for efforts to accurately assess disaster-related mortality and morbidity to guide priorities, improve public health messaging surrounding disasters, and create system-wide efforts that can alter the trajectory of adverse health outcomes amplified by the deleterious effects of disasters. 

## 3. An Urgency to Address the Disproportionate Impact of Disasters on Vulnerable Groups

Over and above the need to understand the short-, and long-term impacts of disasters is the urgency to focus on vulnerable populations. Increasingly, the role of structural factors that drive health-related disparities has become more recognized in both the academic and population literature [21]. Historically, structural factors have been well-recognized in the disaster-focused literature, where the literature from the fields of urban geography [22,23], disaster planning [24], and the social sciences underscores the disproportionate impact of natural disasters on socioeconomically depressed individuals and communities [25,26,27]: they are more likely to be displaced [28], are at increased risk of adverse health outcomes, have less economic opportunity [29], and are less likely to be given access to government funds in the aftermath of natural disasters [30]. 

A key question facing public health officials, emergency management specialists, and post-disaster planners is what conditions can be *modified, changed, or addressed* for vulnerable individuals and populations to limit and prevent climate change-induced adverse health outcomes related to prevalent medical conditions, such as cardiovascular disease, anxiety, and depression, as well as chronic medical conditions, such as diabetes and cancer. 

The outcomes from the impacts of extreme weather events, worsened or ameliorated by one’s economic, health-related, and social capital, are likely to be heterogenous. Therefore, to address these questions, a new, transdisciplinary approach that brings together the disaster resilience and preparedness communities, social scientists, and clinical epidemiologists is urgently needed. To date, several research bodies in the United States have highlighted the importance of employing such an approach, including the National Institutes of Health [31] and the National Academies of Medicine, Engineering, and Science [32]. 

## 4. Challenges to Undertaking Health Research Focused on the Long-Term Impact of Extreme Weather Events

Despite the urgent need to understand and address these consequences, several challenges exist. First, estimating long-term health impacts requires a longitudinal perspective, a strategy which creates methodological challenges to both directly and indirectly attribute the impact of extreme weather events to long-term health trajectories [33]. Although extreme weather events are increasing in both frequency and intensity, they are not predictable enough to build the infrastructure to run rigorous prospective studies. Therefore, studying the long-term impacts is likely to require the use of observational data. Second, stemming from this pragmatic limitation, an understanding of the role played by biological mechanisms in the effect of extreme weather events on health outcomes is hindered by a lack of serial measurements of validated, individual-level biological and psychometric data (e.g., blood pressure readings, depression scale measurements) from before to after the extreme weather event. Last, the current policy environment, particularly in the United States, provides few avenues to proactively address these health impacts at both the individual- and population-level.

Attributing long-term health effects to extreme weather events is a complex task [19]. In the aftermath of natural disasters, a variety of personal, health-related, as well as institutional, social, and economic factors may directly or indirectly affect an individual’s access to healthcare, as well as their sense of security, livelihood, and social networks [34]. Understanding how these factors relate to one another requires careful study by multidisciplinary teams with expertise in sociological theory, social epidemiology, and their application to disaster-related settings—underscoring again the importance of a transdisciplinary approach. 

Although examples exist (often by coincidence) [35], these factors together make the prospective study of the health impacts of climate change unlikely and impractical. As a result, retrospective cohort designs using observational data have been increasingly used as a tool to bridge the knowledge gap. Powerful causal methodological designs (e.g., use of propensity score matching or inverse probability weighting, or quasi-experimental designs, such as differences-in-differences) that limit the effect of unobserved confounding work have been increasingly used in the fields of hurricane and other extreme weather event research to address the limitations of external validity inherent in longitudinal observational studies [36]. 

Similarly, mechanistic, and translational clinical research is often best performed under experimental conditions that are not often practical or ethical in the context of extreme weather events [19]. Such research requires a well-resourced and often purpose-built infrastructure to engage individuals in the study with repeated measurements of various health attributes. Doing so in the context of the general upheaval resulting from extreme weather events, such as hurricanes/cyclones, flooding, or wildfires, requires a careful anticipation of the logistical issues expected in a resource-limited environment, as well as consideration of the ethical implications for assisting disaster victims at times of dire need [37]. 

## 5. A New Research Agenda with Environmental Justice at Its Heart

The challenges outlined are not insurmountable. However, they will require an ongoing commitment, renewed focus, and creative transdisciplinary approaches to study, address, and intervene on the likely disproportionate impact of climate change and climate change-related natural disasters on both the general, and vulnerable population. To this end, we argue that a future research agenda that address climate change-related health impacts needs, at its heart, a focus on environmental justice. This research agenda should be driven by three important considerations, as follows.

### 5.1. A Commitment and Resources for Long-Term Surveillance and Care to Understand Impacts over the Life Course in Individuals and Populations

Current health-related climate change research is underfunded [38]. In the United States, in response to the growing awareness of the importance of climate change, the National Institutes of Health has provided a detailed strategic framework [39] to provide potential investigators with direction. Nonetheless, interest in the health effects of climate change has not been met with the resources to undertake this research.

While in the short-term, the aftermath of extreme weather events presents the clearest opportunity to study the health-related consequences of climate change, to fully evaluate the impact on health, long-term surveillance of impacted populations is required. This is because the impact of natural disasters on populations has the potential to drive epigenetic changes [40] leading to (1) new or earlier expressions of disease, (2) enduring mental health challenges related to stress, and (3) disruption in the continuity of care during the response and early recovery phase—all of which may ultimately cause longer term morbidity and mortality. Furthermore, it is more likely that communities that lack resilient features (e.g., strong community networks, strong tax bases, and political leverage to steer resources during times of emergency) [41] may be more disproportionately affected into the longer term. 

Thus, informed by an environmental justice perspective, understanding the mechanisms and quantifying the disease-specific and overall risk of adverse health outcomes in vulnerable communities will require the commitment of long-term research funds. Precedent exists for these efforts. For example, well-known research cohorts, such as the Framingham Heart Study [42], funded by the NIH since 1948, has examined with considerable rigor the epidemiological nature of cardiovascular disease over the life courses of three generations of individuals. In the age of climate change, this life course approach will be especially critical given the risk of ongoing exposures to multiple climate change-amplified events (for example, hurricanes striking the Gulf Coast) [43].

### 5.2. Emphasis on Intervention: The Use of Implementation Science to Improve Disaster Science

Climate change is likely to make natural disasters, such as hurricanes, more frequent, intense, and destructive [44,45]. Therefore, the role of implementation science to develop, test, evaluate, and improve early warning devices prior to impending natural disasters (particularly hurricanes, wildfires, and heat waves) remains an area of increasing need in the age of climate change [46]. Robust and timely early warning systems provide time to evacuate populations, intensively manage at-risk populations, or plan in times of disasters at the population level. 

However, the investment of time and resources to build these tools will require collaborative and integrated efforts across various stakeholders, including communities at risk, public health bodies, and emergency services. The communication of the impending event to specific individuals requires an effective, timely communicative route with accurate data and a means to alleviate and address the threat, or to reduce exposure to the threat (i.e., the plan of what to do once the risk is communicated). For example, once a wildfire starts and its direction is made clear by forestry and meteorological experts, this information needs to be conveyed to individuals within the fire’s path. The medium (e.g., phone call, text message, door knocking), message (e.g., information to leave, or to stay), and plan (i.e., to evacuate vs to remain in place) needs to be carefully considered. Involved in the process would be various government departments, public health authorities, and emergency management services, all with clear roles and responsibilities. 

Although such disaster and early warning systems exist (e.g., the US Department of Health and Human Services’ emPOWER program), [47] evidence is lacking about efficacy, how best to communicate the risk, and the appropriate messaging to maximize change in individual behavior to avoid natural disaster exposure. Furthermore, to meaningfully undertake these initiatives, rigorous research on *who* is at risk of adverse health outcomes, *what* these adverse health outcomes are, and the subsequent *consequences* of exposure to natural disasters is required. 

A research agenda that uses the tools of implementation science in real-time to study the efficacy of early warning systems (e.g., the RE-AIM framework) [48] will develop new population- and hazard-specific early warning tools that aim to reduce the exposure of those in harm’s way, thus, preventing both short- and long-term mobility, particularly among those who lack the resources to avoid exposure to natural disasters. 

### 5.3. A Transdisciplinary Approach to Addressing Structural Disadvantage—The Root Cause of Environmental Injustice

Structural disadvantage—“an individual’s or a population groups’ condition of being at risk for negative health outcomes through their interface with socioeconomic, political and cultural/normative hierarchies” [49]—underpins the vulnerability of specific populations to environmental hazards. It behooves the research community to study, address, and intervene on these sources of structural disadvantage that heighten the risk of adverse health outcomes from climate change-amplified natural disasters. For example, there is a well-developed consensus of the role that the built environment and lack of green space plays in increasing the risk of heat-related morbidity, particularly in urban settings [50]. In the United States, redlining—the discriminatory policy whereby communities of color were refused mortgages to purchase real estate in particular areas—has been tied to the limited green space development in neighborhoods from where individuals with heat-related morbidity disproportionately reside [51]. These same policies have also been tied to increased risk of hospitalization secondary to air pollution from vehicular congestion, [52] mental illness [53], and increased flood risk [54].

Casting a light on these social forces that currently and subsequently structurally disadvantage specific populations requires a multidisciplinary understanding of legal, political, sociological, and economic frameworks, as well as further downstream health implications that range from healthcare access to subsequent utilization. A key priority is amplifying the voices of those most affected by these environmental injustices to direct change and provide input into possible remedies and pathways forward, using methods, such as community-based participatory research. Thus, developing an understanding of these systemic forces in relation to health outcomes (e.g., the provision of healthcare before, during, and after a natural disaster for insured and uninsured patients) will help determine possible sites of intervention at the system-wide and organizational level, as well as at the individual level. The building of socially conscious urban planning, environmental, and social policies that account for structural disadvantage can then be harnessed to ameliorate the disproportionate impact of climate change on vulnerable communities now and into the future. 

Therefore, to be enduring in its impact, future work that focus on addressing the long-term health effects of climate change and natural disasters will need to follow a team-based scientific approach that captures the salient features that drive environmental health disparities. 

Although this work is not specific to addressing climate change-related morbidity and mortality, the importance of quantifying and intervening to lessen the underlying, interrelated factors that increase health-related risks to society’s most vulnerable remains an important goal in itself. Nonetheless, similar to the unfolding tragedy of COVID-19 and its impact on vulnerable communities, it is these same vulnerable populations that are likely to incur the most deleterious health effects of climate change. 

## 6. Recommended Next Steps

These guiding principles inform the concrete next steps in driving a research agenda focused on the long-term health impacts of climate-amplified natural disasters. To build the necessary capacity to study the impacts of natural disasters in a transdisciplinary fashion, communities of like-minded researchers from different disciplines need to bring together their ideas. The best known US framework that informs health-related climate adaptation is the Centers for Disease Control and Prevention Building Resilience Against Climate Effects (CDC BRACE) framework [55]. The CDC BRACE framework specifically informs research and administrative steps to support climate adaptation strategies that relate to health outcomes. The framework highlights four specific steps, as follows: (1) understanding current climate impacts on health, (2) projecting future impacts, followed by (3) developing and testing potential interventions, and (4) evaluating those interventions designed to limit climate change-related health impacts. Importantly, the framework underscores a transdisciplinary approach by articulating the importance of combining environmental data with epidemiological analyses that can then translate into meaningful public health adaptation strategies of climate-related threats at local, state, and national levels. Grant mechanisms supporting the formation of such local research groups are limited in both scope and support, but they are an essential component of building the necessary infrastructure. In the United States, the NIH [56] and several foundations [57] have started promoting their formation. 

Having formed these transdisciplinary groups of researchers at local levels, creating a network-of-networks that leverages the wide-ranging expertise of disaster-focused researchers across the globe will help bring together a variety of important perspectives, viewpoints, as well as data to build a series of best practices to study the long-term health effects of climate-amplified natural disasters. Such networks are also currently being funded. For example, the US National Science Foundation has commissioned a series of grants that focus on building global climate-focused networks-of-networks to build climate resilience in urban environments [58], bringing together researchers from across the globe from a variety of different academic disciplines. 

## 7. Conclusions

Given the changing climate, future natural disaster events are likely. A concerted effort to mitigate against their deleterious effects on health requires a firm evidence base to inform meaningful policies that limit the trajectory of adverse health outcomes into the longer term. Although this commentary provides insights related to high-income countries (and the United States), these challenges and approaches are equally pertinent to low-income countries that are also grappling with the consequences of climate-amplified natural disasters. Broadly, without doing so and improving on the existing health-related disaster policies, we risk addressing these consequences in a reactive way at greater expense, limiting the opportunity to safeguard communities and vulnerable populations across the globe.

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
