# Peer review of "Closing the Knowledge Gap in the Long-Term Health Effects of Natural Disasters: A Research Agenda for Improving Environmental Justice in the Age of Climate Change"

_ijerph, 2022, doi:10.3390/ijerph192215365_

Round 1

Reviewer 1 Report

This manuscript submitted as a commentary section provides an essential point on the long-term health effects of natural disasters and proposes research agenda. The paper is well written, and the points are clear and reasonable. However, there are some minor points to be discussed. 

  1. Natural disasters in the modern world are often combined with anthropogenic disasters. Hence, some examples of long-term health studies after environmental disasters may be worth reviewing and mentioning, such as the Seveso disaster, the Hebei Spirit oil spill, and the Deepwater Horizon oil spill. 
  2. The paper is mainly based on natural disasters in high-income countries. The absence of the situation in the middle- or low-income countries, in which susceptibility is more widespread and systemic, may be mentioned as describing the limitation of this commentary.

Author Response

Comment

Response

Natural disasters in the modern world are often combined with anthropogenic disasters. Hence, some examples of long-term health studies after environmental disasters may be worth reviewing and mentioning, such as the Seveso disaster, the Hebei Spirit oil spill, and the Deepwater Horizon oil spill.

We appreciate this comment, as a lot of our findings are equally applicable to anthropogenic disasters, several other reviewers informed us that we should focus on climate amplified disasters. Therefore, we have sort to frame our commentary as a climate focused commentary, shifting focus from anthropogenic disasters.

The paper is mainly based on natural disasters in high-income countries. The absence of the situation in the middle- or low-income countries, in which susceptibility is more widespread and systemic, may be mentioned as describing the limitation of this commentary.

We highlight this limitation in the conclusion, while at the same time mentioning that many of our recommendations have relevance to middle-, and low-income countries.

Reviewer 2 Report

Thanks for the opportunity to revise the manuscript “Closing the knowledge gap in the long-term health effects of natural disasters: A research agenda for improving environ-mental justice in the age of climate change” this comment is intended to draw attention to the need to include long-term health effects in natural disasters. Although the topic is important and essential today, I recommend a somewhat broader discussion, with a more grounded section on what is known today about the effects of natural disasters on the long-term human health (or what is not known) - perhaps even a quick review.

In addition, a section just on the effects of climate change on this subject would also greatly increase the value of the article. Climate change affects human health not only through natural disasters, but through increases in temperature and precipitation, which can increase heat and body stress, impacting long term health. I believe this discussion of just climate anomalies and their effects on human health is important to the article.

In summary - many times the article talks about natural disasters and sometimes only about climate change. Broadening the general spectrum to climate change would solve this problem and bring greater strength to the comment. Also, in the challenges part, a section on individual-specific vulnerabilities also needs to be addressed. Many of these diseases cited are the result of genetic load and behavior and have little relation to environmental variables.

Author Response

Although the topic is important and essential today, I recommend a somewhat broader discussion, with a more grounded section on what is known today about the effects of natural disasters on the long-term human health (or what is not known) - perhaps even a quick review.

Thank you for this comment. We have provided an example of the association between hurricanes and cardiovascular disease in the short-, and long-term as an example of what is known about this issue.

In addition, a section just on the effects of climate change on this subject would also greatly increase the value of the article. Climate change affects human health not only through natural disasters, but through increases in temperature and precipitation, which can increase heat and body stress, impacting long term health. I believe this discussion of just climate anomalies and their effects on human health is important to the article.

We appreciate this comment. We have made clear that these long-term health effects caused by natural diseases are linked to climate change.

In summary - many times the article talks about natural disasters and sometimes only about climate change. Broadening the general spectrum to climate change would solve this problem and bring greater strength to the comment. Also, in the challenges part, a section on individual-specific vulnerabilities also needs to be addressed. Many of these diseases cited are the result of genetic load and behavior and have little relation to environmental variables.

We had broadened the discussion to focus on climate amplified natural disasters. Furthermore, we highlight under the second challenge the individual-level metrics that are required to promote study of the additional allostatic load that increases the risk of adverse health outcomes in vulnerable populations after climate amplified natural disasters.

Reviewer 3 Report

Dear Authors,

This manuscript suggested three components of the research agenda to promote environment justice in the age of climate change.

I think the topic is significant, however, I had some difficulties to grasp how to act to respond to damage following natural disasters.

First, authors should add some references which is basis of description at their manuscripts. There are many points which is difficult to distinguish between facts already reported or just authors' opinions.

Second, it is better to add a new part describing concrete future plans.

Lastly, authors mentioned that this topic is common among whole the world. However, cases introduced in this manuscript were focused on disasters occurred in the US. Adding various cases occurred in other places than the US is more helpful to understand that this issue is the global concern.

Author Response

First, authors should add some references which is basis of description at their manuscripts. There are many points which is difficult to distinguish between facts already reported or just authors' opinions.

Thank you for this comment. We have provided added references to substantiate our comments.

Second, it is better to add a new part describing concrete future plans.

We thank the reviewer for this comment. We have elaborated on concrete next steps and some examples of their creation and frameworks in an added section of the commentary

Lastly, authors mentioned that this topic is common among whole the world. However, cases introduced in this manuscript were focused on disasters occurred in the US. Adding various cases occurred in other places than the US is more helpful to understand that this issue is the global concern.

We have added examples of other natural disasters affecting other countries to provide a global context. However, given the authors are based in the US and speak with authority about the limitations of evaluating the long-term effects of climate amplified natural disasters.

Author Response

Comment

Response

The first several paragraphs of the abstract are almost the same as the sentences in the introduction. This

should not be allowed in an academic peer-reviewed journal.

Our understanding is that an abstract is a summary of the manuscript. Some of the lines are similar in order to frame the arguments made in the article. We have amended the abstract to summarize the framing

In section 2, the authors discussed the limited understanding about climate change long-term effects. However,

the motivation to study long-term extreme effects are not well-established. What are the long-term

health effects? Are there any proved facts about it? As explained in section 4, it is still challenging to

attribute long-term health effects to extreme weather events. More discussions about the specific long-term

climate change health effects, whether physical or mental, will be appreciated.

The motivation for studying the long-term health impacts of extreme weather events is discussed in the opening paragraphs. Particularly, we cite concerns about the short-term nature of evaluation and policy of post-disaster health concerns as limiting an understanding of longer term outcomes such as cardiovascular disease (e.g., heart failure, myocardial infarction), identifying populations at risk, and managing those at risk in the longer term.

We provide an example of the impact of hurricanes on cardiovascular outcomes beyond one year as an example.

While we appreciate the ‘push’ from the reviewer, currently there is a lack of evidence to elucidate the longer term health effects of extreme weather events, despite evidence in certain contexts to suggest otherwise. Therefore, the intention of this commentary is to bring attention to this shortfall in the academic literature. 

When talking about environmental injustice, it is hard to separate from other existing structural disadvantages.

As proposed in section 5.3, it needs transdisciplinary efforts to address the environmental injustice.

Are there any priorities when addressing the structural injustices to improve environment injustice? The

authors provide the example of building environment injustice, what are the other environment injustices?

What will be worsen in the climate change age?

The intention is to highlight that current and future environmental injustices are embedded in existing structural disadvantage, through racist practices such as redlining, or how the built environment is constructed in communities with little political power. Environmental injustice is the consequence of structural disadvantage, they are not separate entities. 

The latter questions highlight important deficits in the paper, and are well taken by the authors. We provide further information to address these questions.

Paragraph 1, Line 5: typo ”globalyl”

Thank you for bringing this to our attention. This has been corrected.

Round 2

Reviewer 3 Report

Dear Authors: Thank you for sending the revised version of your manuscript.

I feel that authors did their best to improve the manuscript.

I think that this manuscript is valuable as a paper published in this paper.

Author Response

Thank you

Reviewer 4 Report

Thanks for addressing my comments. I agree to accept the paper in present form. 

Author Response

Thank you